# Bilateral Seminal Vesicle Invasion Is Not Associated with Worse Outcomes in Locally Advanced Prostate Carcinoma

**DOI:** 10.3390/medicina58081057

**Published:** 2022-08-05

**Authors:** Natalia Vidal Crespo, Laura Enguita Arnal, Álvaro Gómez-Ferrer, Argimiro Collado Serra, Juan Manuel Mascarós, Ana Calatrava Fons, Juan Casanova Ramón-Borja, José Rubio Briones, Miguel Ramírez-Backhaus

**Affiliations:** 1Department of Urology, Hospital General Universitario Santa Lucía, 30202 Cartagena, Spain; 2Department of Urology, Hospital Universitario Miguel Servet, 50009 Zaragoza, Spain; 3Department of Urology, Fundación Instituto Valenciano de Oncología, 46009 Valencia, Spain; 4Department of Pathology, Fundación Instituto Valenciano de Oncología, 46009 Valencia, Spain

**Keywords:** prostate cancer, seminal vesicle invasion, radical prostatectomy, biochemical recurrence, survival

## Abstract

*Background and Objectives*: Patients with seminal vesicle invasion (SVI) are a highly heterogeneous group. Prognosis can be affected by many clinical and pathological characteristics. Our aim was to study whether bilateral SVI (bi-SVI) is associated with worse oncological outcomes. *Materials and Methods*: This is an observational retrospective study that included 146 pT3b patients treated with radical prostatectomy (RP). We compared the results between unilateral SVI (uni-SVI) and bi-SVI. The log-rank test and Kaplan–Meier curves were used to compare biochemical recurrence-free survival (BCR), metastasis-free survival (MFS), and additional treatment-free survival. Cox proportional hazard models were used to identify predictors of BCR-free survival, MFS, and additional treatment-free survival. *Results*: 34.93% of patients had bi-SVI. The median follow-up was 46.84 months. No significant differences were seen between the uni-SVI and bi-SVI groups. BCR-free survival at 5 years was 33.31% and 25.65% (*p* = 0.44) for uni-SVI and bi-SVI. MFS at 5 years was 86.03% vs. 75.63% (*p* = 0.1), and additional treatment-free survival was 36.85% vs. 21.93% (*p* = 0.09), respectively. In the multivariate analysis, PSA was related to the development of BCR [HR 1.34 (95%CI: 1.01–1.77); *p* = 0.03] and metastasis [HR 1.83 (95%CI: 1.13–2.98); *p* = 0.02]. BCR was also influenced by lymph node infiltration [HR 2.74 (95%CI: 1.41–5.32); *p* = 0.003]. Additional treatment was performed more frequently in patients with positive margins [HR: 3.50 (95%CI: 1.65–7.44); *p* = 0.001]. *Conclusions*: SVI invasion is an adverse pathology feature, with a widely variable prognosis. In our study, bilateral seminal vesicle invasion did not predict worse outcomes in pT3b patients despite being associated with more undifferentiated tumors.

## 1. Introduction

Seminal vesicle invasion (SVI) is a known adverse pathology feature in radical prostatectomy (RP) specimens. Although SVI incidences have decreased in recent years, especially when compared to the pre-PSA era [1], SVI is still described in around 3.3–4.4% of patients who undergo RP [1,2]. 

SVI is the defining feature of pT3b disease and it confers a higher risk of progression than extracapsular extension alone [3]. However, patients with SVI are a very heterogeneous group, and the prognosis is not only defined by SVI itself. The Gleason grade, PSA value, lymph node involvement, margin status, and lymphovascular invasion are some of the pathological characteristics that may also influence the future behavior of this group of patients [4,5,6,7]. The extent of SVI has also been included in some of the previous studies, with contradictory results about its influence on prognosis [2,4,8,9]. This study aimed to investigate whether bilateral SVI could identify a worse prognosis group compared with unilateral SVI. 

## 2. Materials and Methods

After obtaining Institutional Ethics Committee approval, the *Instituto Valenciano de Oncología* patients’ database was retrospectively reviewed.

Inclusion criteria:

SVI.

Primary RP performed at our Department.

At least two follow-up visits after RP.

Exclusion criteria:

pT4.

Distant bone or visceral metastasis.

Neoadjuvant ADT.

From 2012 to 2020, a total of 152 consecutive patients with SVI (pT3b stage) were found to either have unilateral or bilateral affection. Moreover, 4 of these 152 patients were excluded because pathological data were incomplete due to RP being performed at another center. Two other patients were lost during the initial follow-up. Ultimately, 146 patients were included in the study, according to the inclusion criteria.

### 2.1. Methods

The clinical stage was evaluated according to the 2017 TNM classification [10]. The regional and distant extents of the disease were evaluated using a CT and a bone scan before surgery. The prostate cancer diagnosis was obtained with either a transrectal or transperineal approach as previously described [11,12].

RP was performed by a senior consultant. Pelvic lymph node dissection (LND) was performed when the risk of lymph node invasion (according to the Briganti nomogram [13]) was 5% or higher. A histological examination of the prostate was assessed by a single senior pathologist (ACF). The RP specimens were weighed, measured, stained, and fixed in whole with 10% neutral formalin solution immediately after surgery. Conventionally, 36 tissue blocks containing the apex and the prostate base were examined. Cut tissue slices were stained with hematoxylin and eosin (H & E) using a standard procedure. The pathological stage and histological grading were evaluated according to the 2017 UICC TNM system [10] and the Gleason grading system [14]. Positive surgical margins were recorded when tumor cells were present at the resection lines. SVI was defined as the evidence of prostate cancer cells invading the seminal vesicles’ muscular wall, regardless of the invasion mechanism. The route of invasion was not always documented and, therefore, it was not included in the analysis. 

Follow-up of the patients after RP was performed according to clinical guideline recommendations [15], with a first PSA at six weeks after surgery and then at three months. The periodicity of the follow-up afterward could be modified according to the criteria of the clinician, but the visits were never delayed more than 6 months. Biochemical recurrence was defined as a PSA > 0.2 ng/mL for two consecutive measurements. The decisions for adjuvant or salvage treatments (RT and/or ADT) were made by a multidisciplinary tumor board. Metastasis progression was considered using bone and CT scans, which were both performed at clinician criteria. 

### 2.2. Statistics

For the analysis, we included preoperative variables, such as age, PSA, PSA density, prostate volume, clinical stage, and biopsy ISUP grade. We also included variables related to the prostatectomy specimen, such as margin status, lymph node dissection (LND) features, lymph node infiltration, tumor volume, and ISUP grade. Other postoperative features included: time to biochemical recurrence, time to secondary treatments such as androgen deprivation therapy and/or radiotherapy (either as adjuvant treatment or as a rescue strategy), and time to metastasis and or death (due to prostate cancer or for other causes).

Patient status at the date of the last follow-up was documented, stratifying patients into free of disease, alive with disease, dead due to prostate cancer, or dead for other causes. Patients were censored at the moment of death or the last follow-up. 

Continuous variables were summarized with descriptive statistics (N, median, and interquartile range). Discrete variables were displayed in frequency tables (N, %). Cox proportional hazard models were used for univariate and multivariate analyses. The survival analysis was performed using a Kaplan–Meier method and log-rank test.

The data analysis was carried out using R language programming v. 3.6.3 (The R Foundation for statistical computing, Vienna, Austria). All tests were two-tailed and considered significant when the *p*-value was <0.05.

The primary endpoint was biochemical recurrence-free survival. Secondary endpoints included secondary therapy-free survival and metastasis-free survival.

## 3. Results

During the study period (2012–2018) a total of 146 pT3b patients were identified. Of these patients, 51 (34.93%) presented bi-SVI. The median follow-up for all patients was 46.84 months (IQR 27.35–79.30).

Table 1 shows the clinical and pathological characteristics of the series stratified by uni-SVI and bi-SVI. The median age of the patients at the time of the RP was 64.09 (IQR 56.66–68.74) years. There were no differences between the bi-SVI group and the uni-SVI one for clinical basal characteristics, except for the prostate biopsy results. Among the uni-SVI group, 43.16% of the patients had ISUP grade 1 or 2 on the prostate biopsy, whilst only 29.41% of the patients in the bi-SVI group had it.

Regarding RP specimen characteristics, most of them differ among both groups (see Table 1). Positive margins were most frequently seen in the bi-SVI group (78.43% vs. 58.95%, *p* = 0.029) and tumor volume was higher in this group (9.8 vs. 4.63 mL, *p* < 0.001). Lymph node dissection (LND) was performed in more patients in the bi-SVI group (88.24% vs. 64.21%, *p* = 0.004), and among these, bi-SVI was associated with more pN1 (82.22% vs. 59.02%, *p* = 0.019).

Contrary to all of these, more patients in the uni-SVI group presented with worse ISUP grades in the RP specimens, with 44.21% of the patients having an ISUP grade 4 (20%) or 5 (24.21%). In the bi-SVI group, RP ISUP grade 4 or 5 was present in 21.57% of the patients (*p* < 0.001) (see Table 1).

During follow-up, BCR was seen in 92 patients (63.01%), and metastatic progression in 26 (17.81%). Ninety-eight patients (67.12%) received additional treatment with either radiotherapy (RT), androgen deprivation therapy (ADT), or both. There were nine deaths, four of them due to prostate cancer. Table 2 shows the statuses of patients at the end of the follow-up. No differences were seen among both groups for BCR-free survival (Figure 1), MFS (Figure 2), and additional treatment-free survival (Figure 3). The 5-years BCR-free survival was 33.31% for the uni-SVI group and 25.65% for the bi-SVI group (*p* = 0.44); MFS was 86.03% vs. 75.63% (*p* = 0.1), and the additional treatment-free survival was 36.85% vs. 21.93% (*p* = 0.09).

Univariate analysis showed that BCR was related to PSA [HR 1.005 (95%CI:1.001–1.009); *p* = 0.016], a positive DRE [HR 1.72 (95%CI: 1.10–2.69); *p* = 0.017]; prostate biopsy ISUP grade [HR 2.38 (95%CI: 1.29–3.20); *p* = 0.002]; by lymph node infiltration [HR: 2.89 (95%CI: 1.54–4.42); *p* = 0.001] and by tumor volume [HR: 1.047 (95%CI: 1.019–1.075); *p* = 0.001]. Only PSA [HR 1.34 (95%CI: 1.01–1.77); *p* = 0.03] and lymph node metastasis [HR 2.74 (95%CI: 1.41–5.32); *p* = 0.003] remained significant in the multivariate analysis.

Metastatic progression was influenced in the univariate analysis by PSA [HR: 1.009 (95%CI: 1.001–1.016); *p* = 0.022] and tumor volume [HR: 1.073(95%CI: 1.022–1.126); *p* = 0.005], while only PSA remained as a predictor of metastatic progression in the multivariate analysis [HR 1.83 (95%CI: 1.13–2.98); *p* = 0.02].

Some basal characteristics were associated in the univariate analysis with a higher risk of use of additional treatment, such as PSA [HR: 1.011 (95%CI: 1.005–1.016); *p* < 0.001] and prostate biopsy ISUP grade [HR: 1.98 (95%CI: 1.18–3.32); *p* = 0.009]. Derived from the specimen, margin status [HR: 2.49 (95%CI: 1.41–4.39); *p* = 0.002], lymph node affection [HR: 3.01 (95%CI: 1.47–6.11); *p* = 0.002], and tumor volume [HR: 1.07 (95%CI: 1.03–1.10); *p* < 0.001] were also independently related to secondary treatment. Taken together, the multivariate analysis showed that only a positive margin influenced the future use of secondary treatment [HR: 3.50 (95%CI: 1.65–7.44); *p* = 0.001].

## 4. Discussion

Bilateral affection of seminal vesicles did not influence the future biochemical or metastatic progression of pT3b patients, despite being associated with worse pathological features. Bilateral SVI has been described in 37.6–58.3% of pT3b patients in previous studies [4,8,9,16], so our data are on the lowest limit of that range (34.93%). It is not clear if bi-SVI confers a higher risk of progression for these patients, as the reports have been scarce and contradictory. In the study by Lee et al. [8], they showed a higher risk of BCR among patients with bi-SVI in the multivariate analysis, and these differences were also present when excluding all pN1 patients. Previously, Epstein et al. [4] reported a worse BCR-free survival for bi-SVI patients in their univariate analysis, but this was not confirmed in the multivariate analysis. Similarly, Numbere et al. [9] showed a better progression-free survival for the uni-SVI group, but no differences for cancer-specific mortality. Unlike these, neither Kristiansen et al. [16] nor Badani et al. [17] found differences among patients with uni-SVI and bi-SVI for progression and BCR, respectively. In our study, bi-SVI was not significantly associated with a higher risk of BCR [HR 1.18 (95%CI 0.77–1.91); *p* = 0.44]. We also studied metastatic progression, but again, it was not affected by bi-SVI [HR 1.89 (95%CI 0.86–4.14); *p* = 0.1] There was a higher incidence of both BCR and metastatic progression in the bi-SVI group, despite not reaching statistical significance.

It seems logical to think that bi-SVI would be a predictor of a worse prognosis, as it is associated with worse pathologic characteristics, such as the Gleason score, tumor volume, and lymph node metastasis [8,9]. It is likely that these patients present with more undifferentiated tumors. In our study, bi-SVI was present more frequently in patients with higher prostate biopsy Gleason scores, higher tumor volumes, positive margins, and lymph node affection. These features are considered adverse pathological factors and represent more aggressive and undifferentiated tumors. However, RP specimen GS was higher in the uni-SVI group despite prostate biopsy GS being higher in the bi-SVI group. We could not find any explanation for this incongruence, as it would be logical to associate a more extended tumor, such as bi-SVI, with worse pathological features.

There were some incongruences between presurgical and postsurgical GS for some patients, as 54/143 (37,76%) patients upgraded their GS and 21/143 (14.68%) downgraded their GS. We are not the first ones to report this kind of discordance for high-risk patients [18]. In fact, up to 25% of the patients were initially considered high-risk, undergoing downstaging at the final pathology [17].

Incongruences between prostate biopsy and RP specimen GS can be easily explained by intra- and inter-observer Gleason grading reproducibility [19], but this does not explain why the bi-SVI group had a lesser GS grade in the RP specimen. However, it must be pointed out that most prostate biopsies were performed and analyzed outside our center and by multiple pathologists, while radical prostatectomy specimens were all analyzed by the same pathologist at our center.

Worse pathological characteristics are generally associated with a higher risk of progression and BCR. The BCR-free survival at 5 years was 32.32% for the uni-SVI group and 24.65% for the bi-SVI, but these differences were not significant (*p* = 0.44). Bi-SVI was not associated with a higher risk of BCR in the multivariate analysis. For the population in the study, the 5-years BCR-free survival was 29.47%. These results are in line with those previously reported for pT3b patients (13.9% to 55.8%) [7,8,13,17,18,19].

Prognostic factors for BCR previously reported in the SVI population included GS, pN1, PSA, extracapsular extension (ECE), cT, positive margins, bi-SVI, tumor volume, lymphovascular invasion, and vas deferens affection, with GS being consistently present in almost all the studies, followed by PSA [1,6,7,8,20,21,22,23,24]. In our study, PSA and pN1 were the only predictors for BCR in the multivariate analysis. Contrary to most previous studies, GS was not associated with a higher risk of progression or BCR. An explanation for this could be that in our population, ISUP grades 4–5 was present in only 36.30% of the patients.

Although most pT3b studies have reported BCR results and its predictive factors, MFS is not uniformly described. Pierorazio et al. [1] reported the results of 989 patients with pT3b disease from different eras (pre-PSA era, early-PSA era, and contemporary era). The 5-years and 10-years MFS in the contemporary era were 84.19% and 75.7%, respectively. The clinical stage and RP specimen GS were independent predictors of metastasis development in the multivariate analysis. Swanson et al. [25] reported the results of the SWOG8794 trial [26] SVI subgroup of patients. Patients in this study were randomized to immediate RT after RP vs. observation. The 5-years MFS in the observation group was 72% and 76% in the RT group. At 10 years, it was 47% and 66%, respectively, although these differences were not significant. No specific characteristics were found to be associated with a higher risk of metastasis development in this study. Similar to these previous results, the 5-years MFS for our population was 82.18%. There was no significant difference between bi-SVI and uni-SVI (83.03% vs. 75.63%, *p* = 0.1) for MFS. Only PSA was a significant predictor of metastasis in the multivariate analysis. Previous studies with results for uni-SVI vs. bi-SVI [4,8,9,16] did not report their MFS rates or possible differences, so to our knowledge, this is the first study with such results.

A total of 98 patients required additional treatment after RP, with RT, ADT, or both. Postoperative management of the pT3b patients was not clearly defined. Three RCTs published results favoring adjuvant RT after RP for high-risk patients. The previously mentioned SWOG8794 trial found better BCR-free survival, local recurrence-free survival, and even better MFS and OS in the pT3N0 patients with adjuvant RT [26,27]. However, the subgroup of patients with SVI, MFS, and OS did not reach statistical significance [25]. The other two RCTs, the EORTC 22911 and ARO 96-02, only found differences for BCR and local recurrence, but not regarding MFS, CSS, and OS [28,29]. A recent meta-analysis that included three RCTs comparing adjuvant RT vs. early salvage RT, did not find any differences regarding event-free survival for all the patients included [30]. In the subgroup of pT3b patients, the event-free survival HR was 1.14 (95%CI 0.63–2.04); *p* = 0.67. Thus, it seems that adjuvant RT does not provide a clear benefit for these patients. Observation and salvage RT, if needed, might be a feasible option for these patients.

There is a lack of good quality evidence concerning the use of ADT. Messing et al. [31] reported the results of an RCT comparing immediate goserelin after RP with pN+ vs. deferred treatment. They included 60.2% of patients with SVI. Their results showed better progression-free survival, CSS, and OS in those treated immediately with ADT. Two other RCTs, in this case using flutamide and bicalutamide, failed at finding a benefit for this adjuvant therapy, although we must keep in mind that the blockage in monotherapy with flutamide or bicalutamide was inadequate. In the population of pT3b patients, Siddiqui et al. [32] reported less BCR and local and systemic recurrence for those receiving adjuvant ADT, with no differences for CSS and OS. Similarly, Moschini et al. [22] showed improved BCR-free survival, CSS, and OS for all of the population and in the subset of pT3bN0 patients. For pT3bN1 patients, adjuvant ADT only impacted BCR-free survival. These last two were observational and retrospective studies. The real effect that adjuvant ADT has on these patients remains unknown. It likely influences the time to progression but the effect on survival is not that clear.

Some patients’ characteristics favor the use of further treatment after RP, either used as adjuvant treatment or as salvage treatment. In our study, patients with positive surgical margins were more likely to receive additional treatment after surgery. In the study by Algarra et al. [21], patients with positive margins and those with pN1 were more likely to receive adjuvant ADT. Positive surgical margins were also the only feature associated with a higher risk of receiving RT in the study by Kristiansen et al. [2]. Poelaert et al. [33], reported a higher risk of receiving RT in patients with positive margins and ADT in patients in whom LND was performed. Although there are not many studies regarding the management of pT3b patients, the results seem to be in accordance with ours, with additional treatments in those patients who presented positive surgical margins. This may reflect a tendency to perform adjuvant RT in those R1 patients, but according to the ARMONIC meta-analysis results [30], these patients may not benefit from adjuvant RT. Each patient needs to be individualized and the treatment decision taken in a multidisciplinary context. Nevertheless, the observation seems to be a feasible option for these patients.

Limitations of the present study include its retrospective nature and the fact that it was a single-center study. The population in the study was limited, with a small number of patients included. Moreover, the patients included were highly heterogeneous, including pN1 patients, which may be considered a more advanced or aggressive disease. Differences between the prostate biopsy and RP specimen ISUP grade could also affect patient management and, therefore, be a potential limitation.

The time period included patients referred to our institute during the COVID-19 pandemic. The COVID-19 pandemic reshaped health organizations to cope with the emergency, optimize resources, and minimize the further spread of the infection [34,35,36,37]. Nevertheless, considering this unprecedented clinical scenario, we found no differences in the trend of presentation of such an advanced pathological stage.

Finally, postoperative management of the patients was not standardized; this may have influenced the course of the patients. Definitive conclusions concerning survival or progression-free survival cannot be made. However, this is, to our knowledge, one of the few studies focusing on bilateral SVI and its effect on survival or the future need for additional treatment.

## 5. Conclusions

Bi-SVI was not related to worse oncological outcomes or a higher possibility of requiring further treatment, although it was associated with worse pathological characteristics. We did not find statistically significant differences for BCR-free survival, MFS, or additional treatment-free survival for bi-SVI or uni-SVI.

## Figures and Tables

**Figure 1 medicina-58-01057-f001:**
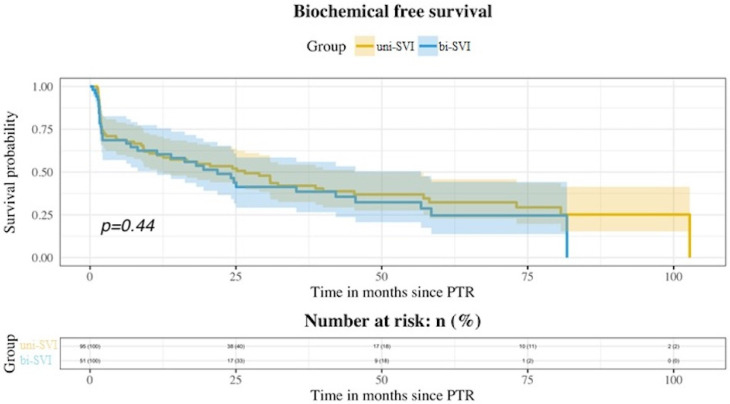
Kaplan–Meier curves showing BCR-free survival for both uni-SVI and bi-SVI groups.

**Figure 2 medicina-58-01057-f002:**
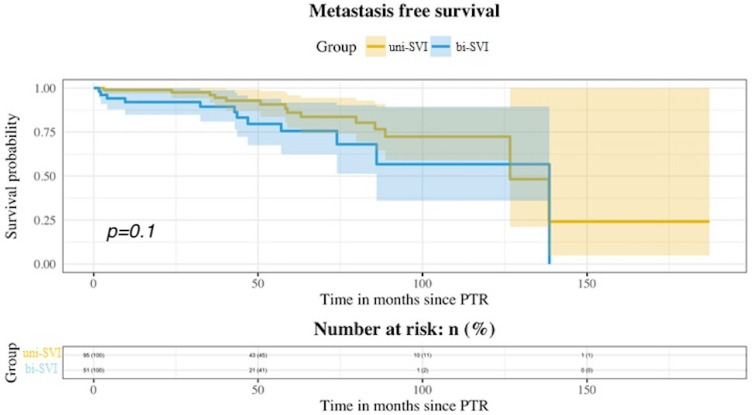
Kaplan–Meier curves showing MFS for both uni-SVI and bi-SVI groups.

**Figure 3 medicina-58-01057-f003:**
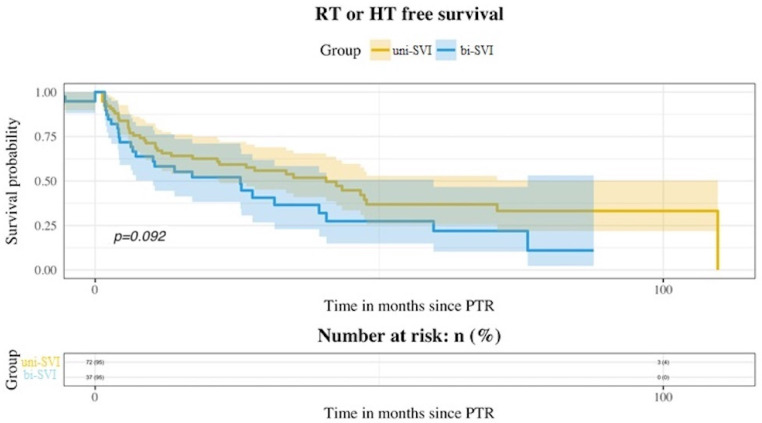
Kaplan–Meier curves showing additional treatment-free survival for both uni-SVI and bi-SVI groups.

**Table 1 medicina-58-01057-t001:** Patient and RP characteristics sorted by uni-SVI or bi-SVI.

Variable		Uni-SVI (95 Patients)	Bi-SVI (51 Patients)	*p* Value
**Age (years) (median and IQR)**		63.89 (59.63–68.45)	64.09 (59.72–68.79)	0.923
**PSA (ng/dL) (median and IQR)**		9.00 (5.98–14.21)	9.06 (7.00–15.56)	0.364
**PSA density (median and IQR)**		0.27 (0.15–0.45)	0.25 (0.17–0.45)	0.647
**Prostate volume (ml) (median and IQR)**		35.00 (26.00–45.00)	40.00 (30.00–46.00)	0.919
**cT**				0.896
	cT1	36 (37.89%)	18 (35.29%)	
	>cT1	59 (62.11%)	33 (64.71%)	
**Prostate biopsy ISUP Grade**				**0.027**
	1	15 (15.79%)	4 (7.84%)	
	2	26 (27.37%)	11 (21.57%)	
	3	21 (22.11%)	19 (37.25%)	
	4	20 (21.05%)	4 (7.84%)	
	5	13 (13.68%)	13 (25.49%)	
**RP ISUP grade**				**0.017**
	2	18 (18.95%)	10 (19.61%)	
	3	35 (36.84%)	30 (58.82%)	
	4	19 (20.00%)	2 (3.92%)	
	5	23 (24.21%)	9 (17.65%)	
**Tumor volume (ml) (median and IQR)**		4.63 (2.80–8.80)	9.80 (5.18–15.88)	**<0.001**
**Positive margin**		56 (58.95%)	40 (78.43%)	**0.029**
**LND performed**		61 (64.21%)	45 (88.24%)	**0.004**
**Number of LN obtained (only patients with LND performed)**		17.00 (11.00–24.00)	18.00 (13.00–24.00)	0.983
**pN1 (only patients with LND performed)**		36 (59.02%)	37 (82.22%)	**0.019**

**Table 2 medicina-58-01057-t002:** Statuses of patients at the end of the follow-up.

	uni-SVI	bi-SVI	All Patients
**Alive and free of disease**	57 (60%)	29 (56.86%)	86 (58.90%)
**Alive with PCa**	32 (33.68%)	19 (37.25%)	51 (34.90%)
**Dead due to PCa**	2 (2.11%)	2 (3.92%)	4 (2.70%)
**Dead for other causes**	4 (4.21%)	1 (1.96%)	5 (3.40%)

## Data Availability

The data presented in this study are available on request from the corresponding author.

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
