# Peer review of "Bilateral Seminal Vesicle Invasion Is Not Associated with Worse Outcomes in Locally Advanced Prostate Carcinoma"

_medicina, 2022, doi:10.3390/medicina58081057_

Round 1

Reviewer 1 Report

Authors performed a retrospective analysis regarding oncologic outcomes in patients with uni- or bilateral invasion of seminal vesicles in patients who underwent radical prostatectomy. They reported that bilateral invasion showed no increased risk for biochemical progression, metastatic events and need for further treatment. This comes in line with studies in the literature although several other studies present conflicting results, probably due to highly heterogeneous populations. My main concern about the study, which is highlighted also by authors, is that a relatively big proportion of patients either upgraded or downgraded disease ISUP grade from transrectal biopsy to final specimen. This is partly explained by authors who claim that biopsies before surgery were performed at different centres, but yet remains a main limitation which should be added also in the limitation section.

Author Response

Dear reviewer, thank you for your revision and the suggested changes. It is true that the differences among the prostate biopsy and the radical prostatectomy ISUP can be a limitation as it can modify the patients’ management. For example, a patient with a lower ISUP on the prostate biopsy than on the RP specimen could benefit from a LND that with the presurgical data would not be indicated and therefore not be performed. Although we already commented it, it is true that it was not stated as a limitation.  We have added this in the limitations section as suggested.

Reviewer 2 Report

In this paper the Authors analyzed the role of bilateral seminal vesicle invasion as a predictor of worse oncological outcomes in prostate cancer. The Authors explored an interesting topic as identifying but also excluding new possible prognostic factors in the era of precision and tailored medicine in prostate cancer is of clinical importance. The manuscript is well structured and written.

Author Response

Dear reviewer, thank you for your revision and your comments. We are glad that you liked our manuscript.